# Development and Replication of a Genome-Wide Polygenic Risk Score for Chronic Back Pain

**DOI:** 10.3390/jpm13060977

**Published:** 2023-06-10

**Authors:** Yakov A. Tsepilov, Elizaveta E. Elgaeva, Arina V. Nostaeva, Roger Compte, Ivan A. Kuznetsov, Lennart C. Karssen, Maxim B. Freidin, Pradeep Suri, Frances M. K. Williams, Yurii S. Aulchenko

**Affiliations:** 1Laboratory of Recombination and Segregation Analysis, Institute of Cytology and Genetics, Novosibirsk 630090, Russia; 2Kurchatov Genomics Center, Institute of Cytology and Genetics, Novosibirsk 630090, Russia; 3Department of Natural Sciences, Novosibirsk State University, Novosibirsk 630090, Russia; avnostaeva@gmail.com; 4Department of Twin Research and Genetic Epidemiology, School of Life Course Sciences, King’s College London, London SE1 7EH, UK; 5Center of Life Sciences, Skolkovo Institute of Science and Technology, Moscow 121205, Russia; 6PolyOmica, 5237 PA s-Hertogenbosch, The Netherlands; 7Department of Biology, School of Biological and Behavioural Sciences, Queen Mary University of London, London E1 4NS, UK; 8Division of Rehabilitation Care Services, VA Puget Sound Health Care System, Seattle, WA 98208, USA; 9Seattle Epidemiologic Research and Information Center, VA Puget Sound Health Care System, Seattle, WA 98208, USA; 10Department of Rehabilitation Medicine, University of Washington, Seattle, WA 98208, USA; 11Clinical Learning, Evidence, and Research (CLEAR) Center, University of Washington, Seattle, WA 98208, USA

**Keywords:** low back pain, GWAS, polygenic risk score, chronic back pain

## Abstract

Chronic back pain (CBP) is a complex heritable trait and a major cause of disability worldwide. We developed and validated a genome-wide polygenic risk score (PRS) for CBP using a large-scale GWAS based on UK Biobank participants of European ancestry (N = 265,000). The PRS showed poor overall predictive ability (AUC = 0.56 and OR = 1.24 per SD, 95% CI: 1.22–1.26), but individuals from the 99th percentile of PRS distribution had a nearly two-fold increased risk of CBP (OR = 1.82, 95% CI: 1.60–2.06). We validated the PRS on an independent TwinsUK sample, obtaining a similar magnitude of effect. The PRS was significantly associated with various ICD-10 and OPCS-4 diagnostic codes, including chronic ischemic heart disease (OR = 1.1, *p*-value = 4.8 × 10^−15^), obesity, metabolism-related traits, spine disorders, disc degeneration, and arthritis-related disorders. PRS and environment interaction analysis with twelve known CBP risk factors revealed no significant results, suggesting that the magnitude of G × E interactions with studied factors is small. The limited predictive ability of the PRS that we developed is likely explained by the complexity, heterogeneity, and polygenicity of CBP, for which sample sizes of a few hundred thousand are insufficient to estimate small genetic effects robustly.

## 1. Introduction

Back pain is a major cause of disability worldwide. According to the Global Burden of Disease Study 2015, low back pain together with neck pain were the leading causes of years lived with disability in 1990–2015 [1]. The most debilitating form of back pain is chronic back pain (CBP; back pain lasting for more than three months). The prevalence of CBP is as high as 30% depending on the population and CBP definition used [2]. It is known that CBP is a complex heritable trait with narrow-sense heritability in the range of 40–68% [3]. Genome-wide association studies (GWASs) using data from national-scale biobanks and electronic medical records have suggested a number of genes to be associated with CBP and similar conditions [4,5,6,7,8,9], with estimated SNP-based heritability up to 13% on the liability scale.

A polygenic risk score (PRS) is an estimate of genetic liability for a trait or disease, calculated according to an individual’s genotype profile and relevant GWASs. A PRS is a powerful modern tool for studying the genetic epidemiology of a disease with potential clinical utility, for example, for developing prognostic biomarkers or subphenotyping patients [10]. There are a few PRS models for general chronic pain [11,12]. These models have been shown to be statistically significantly associated with chronic pain and other traits, but they provide limited ability to predict back pain. Moreover, these models were constructed based on a small number of SNPs or using GWASs with modest sample sizes, and their epidemiological properties have not been sufficiently described. 

In this work, we have developed a genome-wide PRS for CBP based on a recent large-scale CBP GWAS. We assessed the ability of the resulting PRS to predict CBP in an independent sample; examined associations between the PRS and diagnostic codes from the International Statistical Classification of Diseases and Related Health Problems, tenth revision (ICD-10), and Classification of Interventions and Procedures, the fourth edition (OPCS-4), in UK biobank in a hypothesis-free manner; and evaluated the interactions between the PRS and major epidemiologic and clinical risk factors for CBP. 

## 2. Materials and Methods

### 2.1. Developing the PRS Using the UK Biobank

The general scheme of the model creation is presented in Figure 1.

To create and evaluate the CBP PRS model, we used the UK Biobank dataset [13]. Sociodemographic, physical, lifestyle, and health-related characteristics of this cohort have been reported elsewhere [14]. In brief, individuals enrolled in UK Biobank were aged 40–69 years; were less likely to be obese, to smoke, or to drink alcohol; and had fewer self-reported health conditions as compared to the general population. All study participants provided written informed consent, and the study was approved by the North West Multi-Centre for Research Ethics Committee (11/NW/0382). 

Cases and controls of CBP were defined based on questionnaire responses. First, participants responded to “Pain type(s) experienced in the last months” followed by questions inquiring as to whether the specific pain had been present for more than three months. Those who reported back pain lasting more than three months were considered to be CBP cases. Participants who reported no pain or pain lasting for less than three months were categorized as controls, including those who reported pain lasting between 1 month and 3 months. Individuals who preferred not to answer were excluded from the study. We excluded from the analysis individuals who reported more than three months of pain all over the body, because this phenotype was negatively correlated with other pain phenotypes. 

For training the model, we used a CBP GWAS from our previous study based on 265,000 randomly selected individuals of European ancestry (defined using SNP-based principal component analysis) from UK Biobank [4]. A GWAS was carried out using BOLT-LMM v.2.3.2 software [15]. Linear mixed-effect models were fitted to test for additive effects of the SNPs (genotype dosage) on pain phenotypes, adjusting for age, sex, genotyping platform batch, and the first ten principal components of the kinship matrix. Using this GWAS, we developed a set of six draft PRS models using the SBayesR method [16]. The optimal PRS model was chosen based on the maximum area under the receiver–operator curve (AUC) in the UK Biobank validation dataset, which included 30,000 individuals of European ancestry, not overlapping with the 265,000 individuals from the training dataset. Using the optimal PRS model, we calculated PRS values for the rest of the individuals from UK Biobank data using PLINK v 2.0 software [17]. 

For testing the model, we used a set of unrelated individuals (as defined by UK Biobank) of European ancestry (N = 120,217, with 21,543 cases and 98,674 controls) who were not included in the training and validation sets. Prior to testing, PRS values were standardized (transformed to have variance equal to 1 and zero mean). 

### 2.2. Assessment of the CBP PRS Using TwinsUK Data

We then validated the CBP PRS in an independent sample from TwinsUK. TwinsUK is an adult twin registry, predominantly female, which includes over 14,000 participants. It is the most deeply phenotyped and genotyped twin cohort in the world [18]. We used a low back pain (LBP) phenotype as a proxy of CBP. The phenotype was derived from validated self-reported questionnaires [19]. From a subgroup of TwinsUK with available genetic data, we randomly sampled one individual from each twin pair, resulting in 2654 participants in the analysis. Random selection bias was accounted for by running ten permutations of the selection. We used a generalized linear model (GLM) to fit logistic regression using the R stats package (version 4.2.1). Stratified five-fold cross-validation was used to assess the performance of the model. Prior to testing, the PRS values were standardized.

### 2.3. Association of the CBP PRS with Other Health Conditions

We used the PRS of CBP to examine associations with other health conditions, defined by the ICD-10 diagnostic codes (International Statistical Classification of Diseases and Related Health Problems, tenth revision) and OPCS-4 (Classification of Interventions and Procedures, fourth edition) procedure codes, in the set of unrelated individuals of European ancestry from UK Biobank (N = 120,200). We used ICD-10 and OPCS-4 codes combined at the second level (see https://biobank.ctsu.ox.ac.uk/crystal/field.cgi?id=41202, accessed on 30 August 2022 and https://biobank.ndph.ox.ac.uk/ukb/field.cgi?id=41272, accessed on 30 August 2022 for more details on code levels). Codes from the ICD-10 chapters XVIII–XXII and from the OPCS-4 chapters X–Z were excluded. Next, we filtered codes with low prevalence (<0.5% and >99.5%). The resulting list of medical codes contained 165 ICD-10 and 132 OPCS-4 designations. Additionally, we analyzed the association of the PRS with six chronic pain phenotypes (pain in the back, neck/shoulders, hip, knee, stomach, or head, lasting for at least three months) defined by questionnaires, as described above. All traits were analyzed utilizing GLMs and linear regression included in the standard R function glm(). In general, models considered the trait (diagnostic code, procedure code, or chronic pain phenotype) as a dependent variable and the standardized PRS as a predictor with information on sex, age, genotyping batch, and the first 10 principal components (PCs) of the kinship matrix included as the covariates (Trait ~ Age + Sex + batch + PC1 + … + PC10 + PRS). We set the Bonferroni-corrected statistical significance threshold for the analysis at *p*-value < 1.65× 10^−4^ = 0.05/(165 + 132 + 6), where 165 corresponds to the number of ICD-10 codes analyzed, 132 reflects the number of OPCS-4 codes, and 6 represents the number of chronic pain phenotypes.

### 2.4. PRS according to Environment Interactions

We selected twelve known risk factors for CBP (Appendix A). Eleven risk factors were selected based on previous studies: age, sex, BMI, height, physical activity, income (as a proxy of lifestyle), education, alcohol consumption, smoking, depression, and sleep duration [7,20,21,22]. Sitting to standing height ratio was also analyzed, as we have shown that it is associated with CBP [23]. These risk factors were individually tested as covariates interacting with the PRS in a logistic regression model (glm() function). Sex, age, genotyping batch, and the first 10 PCs were included as covariates in all of the models. The significance threshold was set to *p*-value < 0.004 = 0.05/12. 

## 3. Results

### 3.1. Developing the PRS Using the UK Biobank

The optimal PRS model for CBP included 1,090,574 HapMap3 well-imputed SNPs. In the out-of-sample test dataset from the UK Biobank, the mean PRS difference between cases and controls was 0.214 of the PRS standard deviation (two-sided *t*-test *p*-value < 10^−16^). Figure 2A displays the histograms of the PRS distributions for cases and controls. The estimated AUC for CBP was 0.56 (95% CI: 0.557–0.565, Figure 2B). The prevalence of CBP in the test dataset was 18%. The Pearson correlation between PRS and CBP status was 0.082. Using logistic regression, the effect of the PRS on CBP was estimated as 0.22 log odds per PRS SD (OR = 1.24, 95% CI: 1.22–1.26). We estimated the OR of CBP if the individual had PRS from the 80th, 90th, and 99th percentiles of PRS distribution. The ORs were 1.41 (95% CI: 1.36–1.46), 1.49 (95% CI: 1.43–1.56), and 1.82 (95% CI: 1.60–2.06), respectively. 

### 3.2. Assessment of the CBP PRS Using TwinsUK Data

We evaluated the performance of the PRS in an independent TwinsUK sample. The prevalence of LBP in TwinsUK was 28%. The AUC of PRS was 0.55 (95% CI: 0.551–0.559). When we added sex and age into the model using the linear coefficients estimated in UK Biobank (0.216 × PRS + 0.06 × Sex + 0.005 × Age), the AUC increased to 0.58 (95% CI: 0.58–0.59). The effect of the standardized PRS on LBP was comparable with the effect of PRS on CBP in UK Biobank: OR = 1.21, 95% CI: 1.11–1.32.

### 3.3. Association of the CBP PRS with Other Health Conditions and PRS According to Environment Interactions

We evaluated the association of the CBP PRS with ICD-10 and OPCS-4 diagnostic codes (Appendix A). The PRS was significantly associated with 117 codes, with OR point estimates ranging from 1.05 (for S06 “Other excision of lesion of skin”) to 1.35 (for A52 “Therapeutic epidural injection”). Among the strongest associations based on p-values (Table 1), we observed chronic pain at five sites (neck/shoulder, hip, knee, stomach, and headache), M54 “Dorsalgia,” M47 “Spondylosis,” I10 “Essential (primary) hypertension,” and OPCS-4 G45 “ Diagnostic fibreoptic endoscopic examination of upper gastrointestinal tract”.

We evaluated the magnitude of PRS-by-environment interactions with twelve risk factors for CBP. The results are presented in Appendix A. No significant PRS-by-environment interactions were observed after accounting for multiple comparisons, but alcohol consumption and physical activity had nominally significant interaction terms with PRS (*p*-value < 0.05).

## 4. Discussion

Here, we have developed and replicated the first genome-wide polygenic risk score for CBP. The overall predictive ability of the PRS was modest (AUC = 0.56); however, it was highly statistically significant. Moreover, people from the 99th PRS percentile had almost double the risk of CBP (OR = 1.82, 95% CI: 1.60–2.06). Given the high prevalence of CBP, this translates to an absolute risk of 28%. The PRS had almost the same performance and effect estimates in an independent sample from the TwinsUK cohort, and a model also including age and sex (0.216 × PRS + 0.06 × Sex + 0.005 × Age) produced a slightly higher AUC in TwinsUK (0.58). The current work shows that this model is replicable, and suggests that it can be used further on other samples.

An analysis of the associations of the CBP PRS with ICD-10 and OPCS-4 codes showed that the PRS is associated with a wide variety of different diseases and conditions. As expected in line with Cheverud’s conjecture [24,25], the pattern of PRS associations resembles the pattern of genetic correlation of CBP with other diseases [4,9]. Moreover, the PRS for CBP was significantly associated with five other chronic pain syndromes with almost the same magnitude of the effect, in line with previous work, demonstrating that the genetic determination of chronic pain at these locations is shared to a large extent [4,26]. We can determine several groups of traits associated with PRS: coronary artery disease and its risk factors, obesity and metabolism-related traits, traits related to spine disorders and disc degeneration, traits related to diseases of the digestive system, neurological traits (e.g., anxiety disorders), and arthritis-related disorders. These traits are known as potential risk factors for CBP and could be further examined for causal relationships, for example, using Mendelian randomization (MR). Interestingly, we have recently demonstrated a causal relationship between systolic and diastolic blood pressure and CBP [IN PRESS] using MR, which is consistent with the observed association with hypertension seen in the current study. It should be noted that the prevalence of most of the ICD-10 and OPCS-4 codes in UK Biobank is lower than expected in the general population [14]. For example, M54 “Dorsalgia” has only 3% prevalence; however, the same code has 14% prevalence in another population-based biobank—FinnGen (https://r7.risteys.finngen.fi/phenocode/M13_DORSALGIA, accessed on 30 August 2022). Thus, the obtained OR could be underestimated.

The analysis of PRS-by-environment interaction revealed no statistically significant results. This analysis was designed to approximate the analysis of genotype according to environment interactions [10,27]. Nonetheless, we did observe a nominally significant interaction between the PRS and two risk factors (alcohol consumption and physical activity), although the absolute values of the interaction effect were notably small. These results align with our earlier observation that individual SNP × environment interactions are unlikely to have a substantial impact on CBP, and the expected magnitude of the SNP × environment interaction is expected to be negligible for the commonly studied risk factors [28].

The study has several limitations. Firstly, the model was developed using populations of European British descent and should not be generalized to non-European samples. Secondly, the observed squared Pearson correlation of the PRS and CBP was much smaller than the estimated SNP-based heritability on the observed scale for the same trait (0.007 and 0.04, respectively [4]). This suggests that the sample size of the GWAS used for the development of the model (N = 265,000) was not enough to obtain sufficient power. This means that there may be substantial potential for model improvement, and GWASs with a bigger sample size are needed for the development of a more powerful model. With increasing sample size, we could expect that the predictive ability of the model, as well as its potential for clinical utility, will increase.

To conclude, we developed and validated a genome-wide PRS for CBP based on a recent large-scale GWAS, revealing associations with various related disorders and traits, highlighting the challenges posed by the complex nature and polygenicity of CBP.

## Figures and Tables

**Figure 1 jpm-13-00977-f001:**
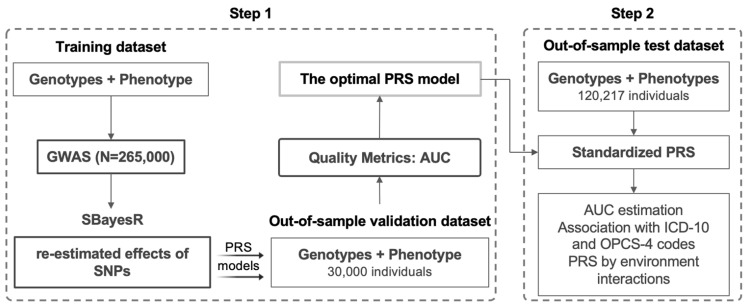
The scheme of the PRS model development and validation in UK Biobank.

**Figure 2 jpm-13-00977-f002:**
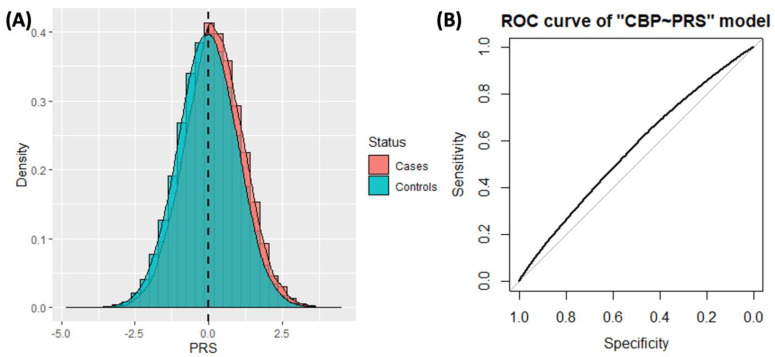
Results of CBP PRS prediction performance. (**A**) Histograms showing the distribution of PRS among CBP cases and controls. (**B**) ROC curve of the CBP~PRS prediction model.

**Table 1 jpm-13-00977-t001:** Top ten associations of CBP PRS with different traits and diseases, ranked by *p*-value.

Prevalence of the Code/Trait, %	*p*-Value	Odds Ratio	Code/Trait Description
17.92	3.56 × 10^−176^	1.24	Back pain for 3+ months
16.17	1.85 × 10^−94^	1.18	Neck pain for 3+ months
17	1.49 × 10^−74^	1.15	Knee pain for 3+ months
9.03	6.65 × 10^−53^	1.17	Hip pain for 3+ months
18.4	3.86 × 10^−51^	1.12	ICD-10 I10 Essential (primary) hypertension
16.76	2.60 × 10^−49^	1.12	OPCS-4 G45 Diagnostic fibreoptic endoscopic examination of upper gastrointestinal tract
3.26	2.96 × 10^−46^	1.26	ICD-10 M54 Dorsalgia
4.85	5.60 × 10^−43^	1.2	Stomach pain for 3+ months
1.72	4.55 × 10^−32^	1.3	ICD-10 M47 Spondylosis
9.21	1.42 × 10^−31^	1.13	Head pain for 3+ months

## Data Availability

The PRS model (PLINK score file) is available on https://mga.icgbio.ru/PRS/Back_3000_genome.csv.gz, accessed on 30 August 2022.

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
