# Peer review of "Development and Replication of a Genome-Wide Polygenic Risk Score for Chronic Back Pain"

_jpm, 2023, doi:10.3390/jpm13060977_

Round 1

Reviewer 1 Report

Thank you for your research.

I read and reviewed the paper with deep interest.

I have attached the review opinion as a file.

Thank you.

Author Response

We would like to thank the Editor and the Reviewers for their helpful comments and suggestions. We have carefully revised the manuscript according to the comments and added clarifications and corrections where necessary. Please find our point-by-point responses to the criticisms raised, below. 

Reviewer 1.

1. I would recommend changing it to chronic low back pain instead of chronic back pain. And recommend CLBP instead of CBP for abbreviations. 

A. We utilized the phenotype definition provided by the UK Biobank questionnaires. While it is highly likely that CBP refers to chronic low back pain, we have maintained the same abbreviation as used in the UK Biobank for accuracy and consistency with our previous publications.

2. [Line 79-80] If one of the subjects has 2 months of pain, is that a control? The criteria for the control group needs to be more clearly presented. 

A: Participants who reported no pain or pain lasting less than three months were categorized as controls, including those who reported pain lasting between 1 month and 3 months. We have added clarification to L79-81 to address this point.

3. I would recommend creating subheadings to separate the results. It will improve readability. 

A.: Thank you for this suggestion, we have adopted this approach. 

4. [Line 208] The reference on physical activity as a risk factor needs to be presented like [209-210].

A.: Yes, we agree, and this is now corrected.

Reviewer 2 Report

1. Please add the present figure regards to the estimated AUC of CBP and AUC of PRS, with e linear coefficients in the paper;

2. Please add a table/figure to show the most valuable associations of the CBP PRS with ICD-10 and OPCS-4 diagnostic codes;

3. Authors have shown a causal relationship of greater alcohol consumption with increased risk of CBP, how about other genotype which did not show association with CBP, please add more discussion about that.

4. Please emphasize the novelty of this study in a conclusion section.

Author Response

We would like to thank the Editor and the Reviewers for their helpful comments and suggestions. We have carefully revised the manuscript according to the comments and added clarifications and corrections where necessary. Please find our point-by-point responses to the criticisms raised, below. 

Reviewer 2.

1. Please add the present figure regards to the estimated AUC of CBP and AUC of PRS, with e linear coefficients in the paper;

A. Thank you for suggesting this clarification. We have included Figure 2, which displays the ROC curve and the distributions of PRS between cases and controls. Additionally, the linear coefficients can be found in line 196 for reference.

2. Please add a table/figure to show the most valuable associations of the CBP PRS with ICD-10 and OPCS-4 diagnostic codes;

A. We have now added Table 1 with most significant associations. 

3. Authors have shown a causal relationship of greater alcohol consumption with increased risk of CBP, how about other genotypes which did not show association with CBP, please add more discussion about that.

A. Thank you for this suggestion. We have revised the paragraph (L230-236) to facilitate a more comprehensive and coherent discussion.

4. Please emphasize the novelty of this study in a conclusion section.

A.: We added a conclusion sentence (L247-249).